# Recurrent *E. coli* Urinary Tract Infections in Nursing Homes: Insight in Sequence Types and Antibiotic Resistance Patterns

**DOI:** 10.3390/antibiotics11111638

**Published:** 2022-11-16

**Authors:** Soemeja Hidad, Boas van der Putten, Robin van Houdt, Caroline Schneeberger, Sacha Daniëlle Kuil

**Affiliations:** 1Department of Medical Microbiology, Amsterdam Infection & Immunity Institute, Amsterdam UMC, University of Amsterdam, Meibergdreef 9, 1105 AZ Amsterdam, The Netherlands; 2Netherlands Reference Laboratory for Bacterial Meningitis, Amsterdam UMC, University of Amsterdam, Meibergdreef 9, 1105 AZ Amsterdam, The Netherlands; 3Department of Medical Microbiology, Amsterdam UMC Location Vrije Universiteit Amsterdam, Amsterdam UMC, University of Amsterdam, Boelelaan 1117, 1081 HV Amsterdam, The Netherlands; 4Center for Infectious Disease Control, National Institute for Public Health and the Environment (RIVM), 3721 MA Bilthoven, The Netherlands

**Keywords:** recurrent urinary tract infection, nursing homes, antibiotic resistance, whole-genome sequencing

## Abstract

Introduction: Recurrent urinary tract infections (rUTI) largely contribute to antibiotic use in older adults. Understanding the genetic characteristics of *Escherichia coli (E.coli)* is needed to identify patients at risk for recurrence. The aim of this study was to obtain a greater understanding of the genetics of *E. coli* rUTI in nursing home residents. Methods: This is a secondary analysis of a multicenter Dutch nursing home study (PROGRESS). *E. coli* strains from residents with a suspected UTI and positive urine culture were analyzed using antimicrobial susceptibility testing and whole-genome sequencing (WGS). Same-strain recurrences were identified by single-nucleotide polymorphism (SNP) analysis. Result: In total, 121 *E. coli* strains were analyzed using WGS, of which 54 belonged to a rUTI episode. One third of *E. coli* rUTI episodes were caused by the same strain (*n* = 18, 33.3%). Same-strain recurrence occurred anywhere between 30 and 434 days after the index UTI, caused by sequence types (ST): ST12, ST23, ST73, ST131, ST453, ST538 and ST2522, in seven nursing home residents. In both single UTI and rUTI, antimicrobial resistance rates were low. Conclusion: Recurrent UTI in nursing home residents are caused by same-strain *E. coli* as well as due to different *E. coli* strains or other uropathogens. Same-strain recurrence can occur over 400 days after the index UTI, suggesting that some strains have the ability to colonize the bladder or gut for longer periods.

## 1. Introduction

Urinary tract infections (UTIs) are one of the most common infections among nursing home residents [1] and contribute to a large burden of disease [2], including mental distress [3]. The majority of UTIs are caused by uropathogenic *Escherichia coli* (UPEC) strains [4,5], which are mostly clonal and belong to the *E. coli* phylogenetic groups B2 or D. The major pandemic lineages globally found are sequence types (ST): ST69, ST73, ST95, and ST131 [6]. Risk factors for recurrent urinary tract infection (rUTI) in older women are urine incontinence, a history of UTI and nonsecretor status. Treatment of rUTI in older adults contributes enormously to the total use of antibiotics [7,8]. The proportion of recurrence in older women varies from 10 to 44% [9,10].

The genetic characteristics of *E. coli* colonizing the intestine in relation to rUTI were previously studied among (younger) adults [9,11,12,13]. Chen et al., characterized *E. coli* strains from stool and urine specimens from women with rUTI, between 18 and 41 years old. Some clonal *E. coli* populations, determined by multi-locus sequence typing (MLST) and whole-genome-sequencing (WGS) analysis, were found in multiple UTI episodes during a 3-month follow-up period, whereas shifts in dominant *E. coli* strains between UTI episodes were also described. It was found that uropathogens were simultaneously present in the urine and the intestine according to clonal tracking, implicating a different potential hypothesis for uropathogen persistence in rUTI (e.g., reinfection of urinary tract from an external source or bacterial persistence within urinary tract) [13]. No distinctive variation was found in the core genome of *E. coli* causing rUTI compared to *E. coli* in non-rUTI based on single nucleotide polymorphism analyses (SNP-analyses) [11]. The current knowledge about the genetics of *E. coli* causing rUTI is mostly based on research in adults instead of older adults (aged ≥ 65 years). One study found no distinctive virulence factors in *E. coli* isolated from rUTI episodes compared to *E. coli* isolated from the index UTI episode by using restriction fragment length polymorphism (RFLP) in women aged 17 to 82 years in primary care [9].

The empirical antibiotic regimen for rUTI is usually based on previous urine cultures and their susceptibility test results, which may help to predict susceptibility [14], as it is known that rUTIs are frequently caused by the re-introduction of the same index strain from the genitorectal area [15,16]. The persistence of infection from internal bladder colonies is an alternative mechanism for recurrence [12,13,16,17]. Most recurrences have been associated with specific antibiotic resistance traits and sequence types, for example: one study found that recurrences of *E. coli* causing UTI were caused by the same phylogenetic group and ST131 subclones [18] and another study reported that ST131 was predominantly observed from patients with rUTI [19].

Increasing antimicrobial resistance hampers the effective treatment of UTIs. As rUTI is a major cause of antimicrobial use and risk for antimicrobial resistance (AMR), strategies are needed to identify patients at risk for recurrence combined with understanding bacterial strain compositions responsible for recurrence to target preventive strategies. The aim of this study is to obtain a greater understanding of the genetics behind rUTI caused by *E. coli* in nursing home residents.

## 2. Materials and Methods

### 2.1. Study Design and Study Population

This is a secondary analysis of the PROGRESS study which assessed the diagnostic accuracy of C-reactive protein and procalcitonin, to diagnose UTI, in a Dutch multicenter study in nursing homes [20]. Nursing home residents (=> 65 years old) residing at psychogeriatric, somatic or rehabilitation wards were consecutively enrolled when provided informed consent and a UTI was suspected by the treating physician or nurse between November 2017 and August 2019. Exclusion criteria were a suspected respiratory tract infection, another infection requiring antibiotic treatment, or previous enrolment in the past 30 days. 

### 2.2. Bacterial Isolates and Definitions

Details on urine specimen collection and bacterial isolation procedures were previously described [21]. Briefly, urine specimens obtained from nursing home residents with a suspected UTI were used for semi-quantitative bacterial culture and urine dipstick testing. The urine culture procedure consisted of 10 μL of urine streaked out on two selective agar plates. After overnight incubation at 37 °C, uropathogens were identified using MALDI-TOF mass spectrometry (Microflex; Bruker Daltonik, Billerica, MA, USA). The identified bacterial strains were classified as uropathogen according to the European Consensus Guideline [22] and antibiotic susceptibility testing was performed when ≥10^4^ colony-forming units per milliliter (CFU/mL) growth was found. The recovered *E. coli* isolates were stored using glycerol peptone 4% at −80 °C until being processed to be used for SNP-analysis (*n* = 135). 

Residents were enrolled when a UTI was suspected according to the treating physician based on clinical signs and symptoms. In this secondary analysis, all residents enrolled with a positive urine culture were considered as having a UTI. Residents with a single UTI episode caused by *E. coli* during the study period were referred to as single UTI. Residents with multiple *E. coli* UTIs (at least a 30-day interval between subsequent UTI episodes, to exclude persistent infections) were defined as *E. coli* rUTI. Same-strain *E. coli* rUTIs were referred to when the same *E. coli* strain was observed over time based on whole-genome sequencing and single-nucleotide polymorphism (SNP) analysis. 

### 2.3. Genome Sequencing, Assembly and Typing

Deoxyribonucleic acid (DNA) was extracted using a QIAamp-DNA mini-kit (Qiagen, Germantown, MD, USA). DNA concentration was measured using a Qubit fluorometer (ThermoFisher, Waltham, MA, USA). Specimens were normalized and KAPA HTP Dual indexed library preparation was performed (Roche, Durban, South Africa). After library preparation, specimens were pooled and sequenced on an Illumina NextSeq platform using the P2 300 cycles kit (Illumina, San Diego, CA, USA). Illumina reads were trimmed using Trimmomatic v0.36 (Bolger 2014), after which reads were assembled using the Shovill v1.1.0 wrapper (https://github.com/tseemann/shovill, accessed on 27 January 2022) for SPAdes v3.13.0 [23]. Isolates were excluded if the genome size was outside the range of 4.5–5.6 Mbp, the N50 was lower than 30,000 bp, or the assembly consisted of more than 500 contigs, all assessed using Quast v4.6.3 [24]. Sequence types were inferred using mlst v2.19.0 (https://github.com/tseemann/mlst, accessed on 27 January 2022). *fumC*-*fimH* clonotypes were determined using the CHtyper webtool [25], serotypes were predicted in silico using ECtyper v1.0.0 [26], and EzClermont v0.6.3 was used to infer phylogroups [27].

### 2.4. Analysis of Strain Recurrence by SNP Analysis (Same-Strain rUTI)

Isolates were included in the SNP analysis if multiple isolates of the same sequence type were identified in the dataset. SNP analysis was performed using snippy v4.4.5 (https://github.com/tseemann/snippy, accessed on 27 January 2022), with reference genomes selected per sequence type using reference seeker v1.8.0 [28]. We defined the recurrence of the same *E. coli* strains (same-strain rUTI) as follows: (i) ≥2 *E. coli* strains isolated from the same participant; (ii) from different UTI episodes (positive urine culture obtained at different time points with at least 30 days between sampling); (iii) during the study period. *E. coli* isolates are considered ‘the same strain’ when the SNP differences are 25 or less [29]. A phylogenetic tree of all strains was constructed by mapping sequence reads on the ATCC 25922 reference genome (CP009072.1) as described above.

### 2.5. Antibiotic Resistance Rates 

Antimicrobial susceptibility testing was performed by using theVITEK2 platform (BioMérieux). Results from the most recent *E. coli* strain in time were analyzed. AMR rates were calculated based on available susceptibility data. 

### 2.6. Descriptive Analysis

Descriptive statistics were reported using Social Sciences software (SPSS) for Windows version 17.0 [30]. Figures and tables were constructed using Microsoft Excel and CANVA [31,32].

## 3. Results

The present study is based on a dataset from a multicenter nursing home study (PROGRESS) with the broader aim to improve UTI diagnosis [21]. The dataset consisted of 298 suspected UTI episodes based on clinical signs and symptoms (208 unique nursing home residents): 149 single suspected UTI episodes (71.6%) and 59 nursing home residents with suspected rUTIs (*n* = 149 episodes, 28.3%). Of these 149 suspected rUTIs, 69 episodes (39 unique nursing home residents) were urine-culture-positive for growing *E. coli*, further referred to as *E. coli* rUTI (46.3%). Due to missing isolates and/or technical errors, 54 out of 69 *E. coli* episodes (78.3%) were included for SNP analysis. Another 14 strains were excluded due to technical errors/missing information (*n* = 7), or novel sequence types observed from SNP analysis (*n* = 7). Of the remaining 121 *E. coli* isolates, about half were single UTIs (*n* = 67, 55.4%), the remaining were rUTIs (*n* = 54, 44.6%); see Figure 1.

### 3.1. E. coli Sequence Types in Nursing Home Residents with Single Versus Recurrent UTI

Overall, the most frequently isolated *E. coli* sequence types were ST131 (*n* = 20, 16.5%), ST73 (*n* = 16, 13.2%), ST69 (*n* = 12, 9.9%), ST10 (*n* = 6, 5.0%), ST12 (*n* = 5, 4.1%), ST38 (*n* = 5, 4.1%), ST362 (*n* = 3, 2.5%), ST405 (*n* = 3, 2.5%), ST453 (*n* = 3, 2.5%) and ST538 (*n* = 3, 2.5%). These ten sequence types represented more than 60% of all included *E. coli* isolates. 

ST69 was more frequently isolated among individuals with a single UTI compared to individuals with rUTI (16.4% versus 1.9%) whereas ST73 and ST12 were more often found among individuals with rUTI (18.5% versus 9.0% for ST73; 9.3% versus 0.0% for ST12); see Table 1 and Figure 2.

### 3.2. Relatedness of E. coli Isolates Based on SNP Analysis (Same-Strain Recurrence) from Seven Nursing Home Residents

Of the 54 *E. coli* rUTI episodes, 18 episodes (seven unique nursing home residents) were identified with at least two episodes caused by the same-strain *E. coli* rUTI based on SNP analysis (33.3%); see Figure 3 (the same color reflects the same strain). When plotted in a phylogenetic tree, most UTI episodes (recurrent and non-recurrent) belonged to phylogroups B2 and D of *E. coli* (see Appendix A). We could not observe a clear association between recurrent UTI and phylogenetic placement.

Genotypical results were not available for three episodes. Recurrence occurred anywhere between 30 and 434 days after the index UTI (see Figure 3). The same-strain rUTI occurred late in resident 5 (377 days after index UTI episode) and resident 6 (399 days after index UTI episode). The signs and symptoms in rUTIs with the same strain were sometimes similar, but in other same-strain rUTIs, different signs and symptoms were observed. The initial presenting symptoms improved during follow-up (day 5 or 10) for most episodes (*n* = 16, 88.9%).

Overall, seven different *E. coli* sequence types (ST2522, ST23, ST453, ST538, ST12, ST73 and ST131) were identified with less than 25 SNPs difference between *E. coli* isolates within an individual nursing home resident, which indicates recurrence by the same index strain. In one nursing home resident (see Figure 3, line 3), two rUTIs were identified: the first rUTI episodes caused by ST23, and the subsequent rUTI episodes caused by ST73. The majority of same-strain *E. coli* UTI episodes were treated with antibiotics (80%, *n* = 16) and all but one were treated with nitrofurantoin (93.8%, *n* = 15). The duration of treatment is unknown, but the national UTI guideline for frail elderly recommends a treatment duration of 5 days [33]. There were no nitrofurantoin-resistant *E. coli* isolates found among the same-stain *E.coli* causing UTI. The overall nitrofurantoin resistance for *E. coli* was low (4.1% in the total dataset, 6/148, Appendix A) and there were no UTIs caused by (intrinsically) nitrofurantoin-resistant uropathogen (e.g., *Proteus* spp.) after previous nitrofurantoin antibiotic treatment of the index UTI episode.

### 3.3. Antibiotic Resistance in E. coli Causing rUTI

The antimicrobial resistance rates in *E. coli* rUTI (both same-strain and non-same-strain based on SNP analysis) (*n* = 49 episodes from both same-strain and non-same-strain *E. coli* with susceptibility data available) were: 44.1% for amoxicillin, 42.1% for amoxicillin–clavulanic acid, 13.1% for ciprofloxacin, 10,7% for sulfamethoxazole–trimethoprim, 2,6% for fosfomycin, and 2.6% resistance against nitrofurantoin. The overall antibiotic resistance rates are listed in Appendix A.

## 4. Discussion

In this secondary analysis from a multicenter study conducted in nursing home residents, we focused on rUTIs caused by *E. coli*. We showed that rUTI episodes were caused by other *E. coli* strains (sequence types) and *E. coli* strains unique (≤25 SNPs difference) to the index strain (same-strain) determined by whole-genome sequencing. The time between the index UTI and subsequent UTI episodes caused by the same *E. coli* strain varied substantially between nursing home residents. Interestingly, the longest time between an index UTI episode and rUTI episode was 434 days. It is known that a shorter timeframe between two UTI episodes increases the likelihood of an infection by the same strain [9,34,35]. Our findings on same-strain recurrence after >400 days suggest that *E. coli* is able to either colonize the bladder for very long periods or recolonize the bladder through the re-introduction of *E. coli* strain from the gut, which is currently not well understood. It remains unknown why some residents suffer from recurrences by the same strain, while others are re-infected with a new strain. It could be hypothesized that treatment with antimicrobials with tissue penetration (unlike nitrofurantoin) eliminates the causing *E. coli* strain from both the bladder and the genitorectal area during treatment, which may lead to subsequent infections with other strains, instead of same-strain recurrences. We could not support this hypothesis as most residents were treated with nitrofurantoin and only a few nursing home residents were treated with antimicrobials allowing tissue penetration.

As uropathogenic *E. coli* acts as a reservoir for the development and mobilization of novel resistance genes or combinations of resistance genes in the gut and at infected extraintestinal body locations [36], it is necessary to consider gut–bladder transmission of these strains as a pathway for UTIs. Unfortunately, we did not collect any stool samples and were therefore unable to study gut–bladder transmission.

The sequence types ST131 and ST73 were most commonly found which is in line with previous data [6,36] and we found comparable *E. coli* sequence types in nursing home residents with a single UTI and rUTI (see Appendix A
Appendix A) [11].

Overall AMR rates were low for the five most commonly prescribed antimicrobial agents (amoxicillin–clavulanic acid, sulfamethoxazole–trimethoprim, ciprofloxacin, fosfomycin and nitrofurantoin) and the most identified *E. coli* strains were susceptible to nitrofurantoin in both rUTI and single-episode UTIs such as previously described in rUTI [37,38].

Nitrofurantoin is the recommended empirical antibiotic therapy for uncomplicated UTIs in the Netherlands and was therefore relatively frequently prescribed. The low nitrofurantoin prevalence and absence of selection of (intrinsically) nitrofurantoin-resistant uropathogens in time suggest that nitrofurantoin susceptibility isolated from an index urine culture might predict nitrofurantoin susceptibility for future UTI episodes in this setting. However, this hypothesis should be tested using a sufficient sample size to minimize bias introduced by selection or low numbers included.

To the best of our knowledge, this is the first study evaluating *E. coli* sequence types in relation to rUTI in nursing home residents. Due to the study design of the primary study (PROGRESS study), nursing home residents were eligible to enroll multiple times which enabled comparison of UTI episodes within residents including residents living at psychogeriatric wards. This group is less frequently studied due to logistic challenges and ethical concerns, but are a particular risk group for rUTI. In addition, we enrolled both residents who were treated with antibiotics and those who were not treated, which closely mimics clinical practice. Moreover, using molecular techniques to identify sequence types gives more in-depth information on *E. coli* strain composition compared to commonly used phylogenetic analysis by PCR. This created a unique opportunity to study the occurrence and patterns of *E. coli* causing rUTI over time.

However, there are some limitations in this study. First, misclassification bias (differential misclassification) could have been introduced when a subsequent UTI episode was not included due to various reasons such as: UTIs prior the start of the study, referral or death of the nursing home resident during the study period, or the nursing home resident not applying for enrolment. This was caused by the secondary analysis while the data were collected for a different purpose. This may have led to an overestimation of single UTI episodes in our study. For this reason, we are unable to make any firm statements about differences between nursing home residents with rUTI and nursing home residents with single UTI.

Second, a subset of available *E. coli* isolates was sequenced, while one third of the (recurrent) UTI episodes were caused by other uropathogens (e.g., *Klebsiella* spp., *Proteus* spp. and *Aerococcus* spp.). Therefore, we cannot make any conclusions about rUTIs caused by other uropathogens. Third, from all isolated *E. coli* strains, a single colony was used for SNP analysis. There could be some variation within the host or the existence of different *E. coli* clones [39] which may lead to an underestimation of species diversity.

Although many studies focus on unraveling the mechanism behind rUTI, the landscape remains unclear. Future studies should target specific genetic characteristics (for, e.g., virulence factors) which may help in predicting recurrence and helps to target preventive strategies. Preventive intervention strategies such as immunoprophylaxis could be informed by the genetic information of recurrent strains, and may reduce the burden of disease caused by pathogenic and resistant *E. coli* strains [40].

## 5. Conclusions

Recurrent UTIs in nursing home residents are caused by same-strain *E. coli* as well as due to different *E. coli* strains or other uropathogens. Recurrence by same-strain *E. coli* occurs after several months up to over 400 days after the index UTI episode, which suggest that some strains have the ability to colonize the bladder for very long periods or recolonize the bladder by the re-introduction of the *E. coli* strain from the gut.

## Figures and Tables

**Figure 1 antibiotics-11-01638-f001:**
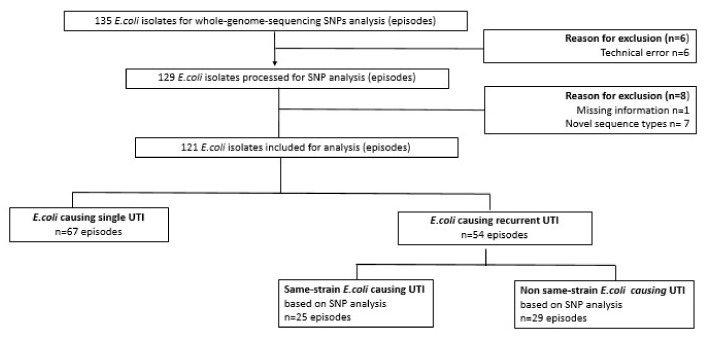
Overview of Escherichia coli (*E. coli)* isolates analyzed by whole-genome sequencing. *E. coli* causing single urinary tract infection (UTI) was defined as a single UTI episode with ≥10^4^ CFU/mL growth of *E. coli* in the urine culture; *E. coli* causing recurrent UTI (rUTI) was defined as unique nursing home residents with ≥2 UTI episodes with ≥10^4^ CFU/mL growth of *E. coli* in the urine culture at ≥30 days’ interval between subsequent episodes; same-strain *E. coli* causing UTI was defined as the same *E. coli* strain observed over different time points based on whole-genome sequencing and single-nucleotide-polymorphism (SNP) analysis; non-same-strain *E. coli* was defined by a subgroup of nursing home residents with *E. coli* causing rUTI but not same-strain based on SNP analysis.

**Figure 2 antibiotics-11-01638-f002:**
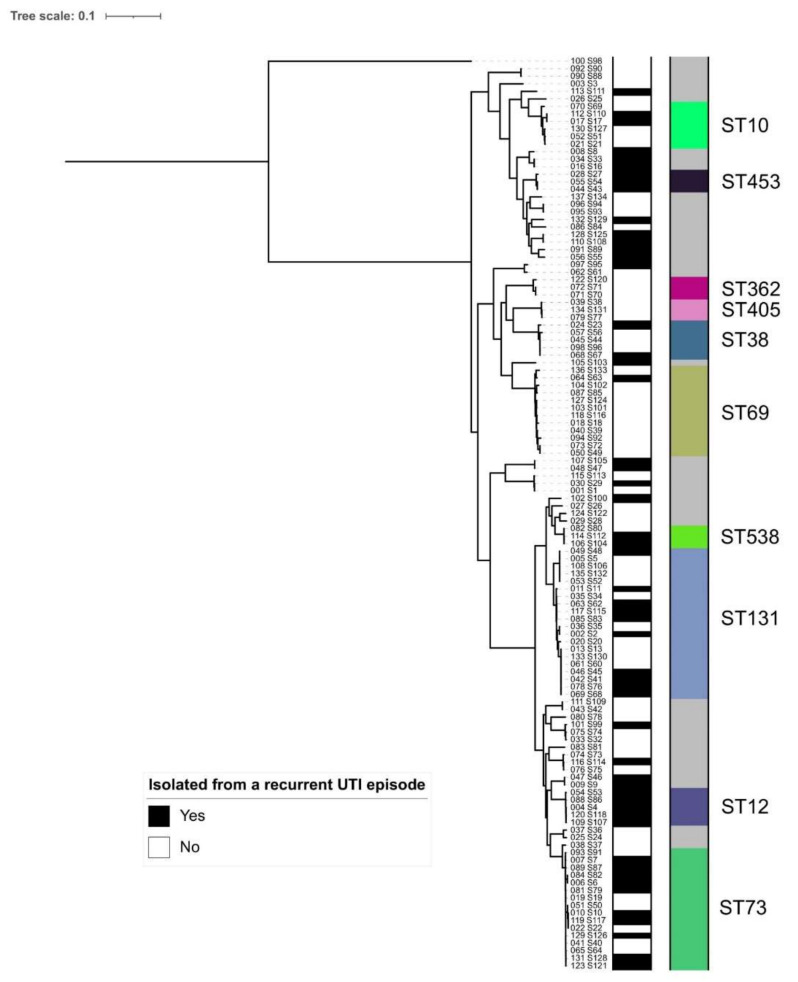
Phylogenetic tree inferred from SNPs from a whole-genome alignment for all sequenced *E. coli* strains. The left bar indicates whether an isolate was sampled from a rUTI episode (black: yes; white: no) and right bar indicates the ten most common sequence types, as listed in Table 1.

**Figure 3 antibiotics-11-01638-f003:**
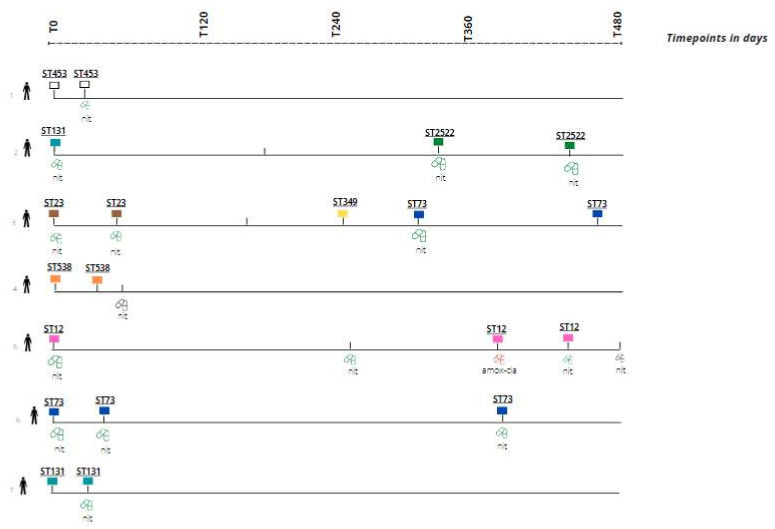
Overview of nursing home residents (7 unique individuals) with recurrent UTI caused by same-strain *E. coli* identified based on SNP analysis. Box color represents *E. coli* sequence type identified; the same color reflects the same strain. STxxx represents sequence type. No box indicates: no urine culture result/isolate available. Green pill indicates a susceptible *E. coli* strain for antibiotic treatment prescribed; red pill indicates resistant *E. coli* strains for antibiotic treatment prescribed. Black pill indicates antibiotic treatment with lacking susceptibility testing results. amox/cla = amoxicillin–clavulanic acid; nit = nitrofurantoin; ST = sequence type; T0 = day of the index UT.

**Table 1 antibiotics-11-01638-t001:** Frequencies of *E. coli* sequence types isolated from Dutch nursing home residents. Detailed typing is listed in Appendix A.

Sequence Type	Overall (*n* = 121)	Recurrent UTI (*n* = 54)	Non-Recurrent UTI (*n* = 67)
		Number of episodes (%)	
131	20 (16.5)	10 (18.5)	10 (14.9)
73	16 (13.2)	10 (18.5)	6 (9.0)
69	12 (9.9)	1 (1.9)	11 (16.4)
10	6 (5.0)	2 (3.7)	4 (6.0)
12	5 (4.1)	5 (9.3)	0 (0.0)
38	5 (4.1)	2 (3.7)	3 (4.5)
362	3 (2.5)	0 (0.0)	3 (4.5)
405	3 (2.5)	0 (0.0)	3 (4.5)
453	3 (2.5)	3 (5.6)	0 (0.0)
538	3 (2.5)	2 (3.7)	1 (1.5)
Other *	45 (37.2)	19 (35.1)	26 (38.7)

* other sequence types identified: ST2, ST14, ST23. ST58, ST59, ST88, ST93, ST104, ST127, ST141, ST345, ST349, ST357, ST362, ST404, ST415, ST428, ST550, ST636, ST646, ST648, ST847, ST998, ST1236, ST1300, ST1444, ST1771, ST1844, ST2015, ST2017, ST2280, ST2522, ST2914, ST3236, ST6467, ST7092.

## Data Availability

The data in this study are available from the corresponding author upon request.

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
