# Peer review of "Recurrent E. coli Urinary Tract Infections in Nursing Homes: Insight in Sequence Types and Antibiotic Resistance Patterns"

_antibiotics, 2022, doi:10.3390/antibiotics11111638_

Round 1
Reviewer 1 Report
This study aimed to obtain a greater understanding of the genetics of E. coli recurrent urinary tract infections in nursing home residents.
The report is well-written and provides up-to-date data on AMR to the public health community.
Taken together, the paper could be accepted in its current form.
Author Response
Please see the attachment.
Thank you for your review.

Reviewer 2 Report
A small list of pathogens found during the study or in Ecuador, in general, would be interesting.
A short discussion about the low SSI number would improve the paper's quality.
In this paper, titled “Recurrent E. coli urinary tract infections in nursing homes: insight into sequence types and antibiotic resistance patterns”. Hidad et al, aim to study E. coli sequences in psychogeriatric and nursing home residents in relation to rUTI. Some obvious limitations of this study are clearly highlighted. This work is interesting and well-written but needs to answer the following points. In table 1, in the legend, the word “serotype” is used. For me there is a mistake as only isolate types are in the table. Data about the serotypes should be added. What about the subclone, clonotype, and clade of each isolate or at least of the recurrent cases? A short discussion about the in vivo and in vitro evolution experiments of E. coli could increase the interest of the readers.
Reviewer 3 Report
This is an interesting study about recurrent UTI. However, it could be improved :
1-line 111 : the definition of rUTI is not clear : what is multiple UTI? Moreover, in line 131-132, a 30-days interval was mentioned, is that included or not in the definition of rUTI, if so it must be mentioned before in line 11A
2- table 1 : the p-value must be added in the table to compare the frequencies, also in the text. No statistics were shown in the results but mentioned in the materials
3- there are contradictions in the % of nitrofurantoin resistance: line 228 "There were no nitrofurantoin resistant E. coli isolates found. " but in line 238 "2.6% resistance against nitrofurantoin " , please check the % of furane resistance and it would be preferable to precise the % of furane resistance in simple UTI and rUTI and compare them statistically ( provide p value).
Reviewer 4 Report
In the presented manuscript, Hidad and co-workers analyzed “Recurrent E. coli urinary tract infections in nursing homes: insight in sequence types and antibiotic resistance patterns”.
The manuscript is suitable for publication in Antibiotics, the language should be revised and additional analyses should be performed to present a comprehensive analysis as proposed by the authors.
I provide some major and minor comments below.
Major;
The authors mention composition of the E. coli strains, and an unique opportunity to study the occurrence and patterns, however to compare the mentioned data it is necessary to perform the following:
1. It is imperative that the authors provide a phylogenetic tree where they include the strains analyzed by SNP indicating which are recurrent and which are not.
2. The authors should perform a MLST phylogenetic tree to compare the results shown and also compare them with the SNP phylogenetic tree.
3. In the manuscript it is difficult to follow E. coli strains unique and other E. coli strains causing recurrence, a sankey plot where these data are included as well as time points in days, susceptibility and antibiotics would facilitate understanding for the reader.
4. The authors do not report the size, coverage and quality of the genomes used in the study.
Minor,
The writing and grammar of the article must be revised.
Abstract
Define in the summary which strain is causing the UTI recurrences. As well as the other strains that also cause recurrences and the other uropathogens.
1. Line 25: Escherichia coli (E. coli). The scientific names in italics.
2. Line 63-64: How do the authors define a distinctive variation in the genome?
3. Line 65: Little is known about the genetics of E. coli causing rUTI in older adults. Rewrite this sentence; there are hundreds of articles that analyze this problem.
4. Line 79: AMR, please define.
5. Line 115-121: The genome assembly method is not provided. It is provided in the next section that does not correspond.
Supplementary materials S1: Overview sequence types
Escherichia coli. The scientific names in italics.
Round 2
Reviewer 4 Report
Minor,
The writing and grammar of the article must be revised.
1. Line 25: Escherichia coli (E. coli). Delete.
Check the scientific names in the references; the authors do not present them in italics.
Author Response
Thank you for your comments. We have revised all questions.